# Changes in Haredi Education in Israel: A Comparative Perspective from the United States Using Monsey as a Test Case

Ilan Fuchs 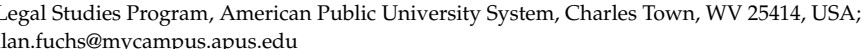

Legal Studies Program, American Public University System, Charles Town, WV 25414, USA;
ilan.fuchs@mycampus.apus.edu

**Abstract:** The Haredi community in Israel is the youngest community. It grows at a higher rate than that of the Arab population in Israel. The calls to introduce more secular education are motivated by both a desire to acculturate the Haredi population and bring it closer to the norms and values of the Israeli discourse and from a wish to integrate more people from this community into the job market. Calls to introduce more secular education in the Haredi system have seen constant resistance that has been documented in scholarship. In the U.S., the discussion on the correct role of general education started with a different frame of reference. The existence of meaningful religious education was put into question, and only after the holocaust did American Orthodoxy significantly expand its educational options. This text will describe the diverse spectrum of Haredi educational institutions and their approaches to secular education. I am using Monsey as a test case since it is a good representation of the kaleidoscope of Orthodoxy in the U.S., as far as the different communities are concerned. It is also a very decentralized community, since there is no dominant group in Monsey. This fact allows for more initiatives, including educational initiatives, to be undertaken.

**Keywords:** Jewish education; Haredi society; Monsey; comparative approach

## 1. Introduction

Israel is a significant regional player in the Middle East. The future of Israel will have major effects on the region in its entirely. Discussion about the future of the Israeli economy, scientific advances, and military development are obviously extremely important. However, demography is as important, if not more so. There is a substantive body of scholarship concerning the demographic changes in the Jewish and Palestinian populations in Israel, Judea, and Samaria. But that is not the only point of interest in Israeli demography. In recent years, we have seen a growing scholarly interest in quantitative developments concerning one segment of the Jewish population, namely the Haredi community.[1] The future of this community and its role in Israeli society has been a topic of interest to many scholars and a recurring theme in the scholarship on the issue was the limited and many times absence of general education curriculum.

The Haredi community in Israel is the youngest community. It grows at a higher rate than that of the Arab population in Israel. The calls to introduce more secular education are motivated both by a desire to acculturate the Haredi population and bring it closer to the norms and values of the Israeli discourse and by a wish to integrate more people from this community into the job market. Calls to introduce more secular education in the Haredi system have seen constant resistance, which has been documented in the scholarship. In general, this resistance is explained by the perception of the introduction of secular education as an attempt by the secular establishment to introduce foreign and secular—namely anti-religious—content and values to the Haredi community. This discourse states that general education is not neutral as far as values are concerned. Suggestions to include general education are dismissed as an attempt to secularize Haredim from within. The definition of Haredi is not as clear-cut as it used to be. The growth in this sector created

more subsections that differ from one another. For the scope of this article, I define the term Haredi as referring to Orthodox Jews who view modernity critically and reject Zionism as a secular national movement. As a part of this rejection of Zionism and the legitimacy of secular Jewish nationalism, members of the Haredi community do not, by and large, serve in the Israeli military.[2]

In the U.S., the discussion on the correct role of general education started from a different frame of reference. The existence of meaningful religious education was put into question and only after the holocaust did American Orthodoxy significantly expand its educational options. This text will describe the diverse spectrum of Haredi educational institutions and their approaches to secular education. I am using educational institutions located in Monsey as a test case, since it is a good representation of the kaleidoscope of Orthodoxy in the U.S., as far as the different communities are concerned. It is also a very decentralized community, since there is no dominant group in Monsey. This fact allows for more initiatives, including educational initiatives, to be undertaken. The educational discourse in Monsey, with its different versions of Orthodox communities, can show how different models of secular education have emerged in similar economic, political, and sociological conditions to fulfill the needs of different communities on the Orthodox spectrum.

The results show that the American Jewish–Orthodox spectrum has created several models for the integration of secular education. These models all have something in common: they view general education as utilitarian and vocational in nature and reject the ethos of liberal arts education. That being said, this rhetoric is not uncommon in the U.S. and is becoming more popular with the changing landscape of higher education in the U.S. American Orthodox models can serve as a useful template for initiatives in Israel where economic questions are part of political discussions concerning Haredi education.

## 2. Educational Trends in the Haredi Community in Israel

The Haredi community is faces with poverty; it has been facing poverty from its inception,[3] and while there has been a decline in the rate of poverty, it is still very high. In 2020, 42% of Haredi families were categorized as being under the poverty line.[4] This decline is an obvious example of the change that this society is going through: a transformation of its approach to Torah study, economy, and, more importantly, its relationship with the secular world and its values.

Scholars of the Haredi community agree that the questions of education are critical for the future of the community but also to that of Israel in its entirely, considering the projections that, by 2065, about 40 percent of Israeli Jews will be Haredi.[5] Projections for the distant future should be considered cautiously; however, they do not change the fact that the characteristics of Haredi education have enormous influence, not only on religious culture, but also on the economy of the Haredi community. Even if we assume that there will be a decline in the growth of the community, in recent years, Haredi has been the community with the highest birthrate in Israel, according to the Israeli Central Bureau of Statistics. Since the beginning of the decade, the birthrate among Haredi women has stabilized at 6.6 children per woman, a slight decline from a little over 7 in the previous decade.[6] Even though there has been a rise in the birthrate in all segments of the Jewish population, from secular to religious, there is no doubt that we can assume two things. First, the Haredi community will continue to be a young population. Second, this population will play a growing role in Israeli society; its options concerning participation in the economic structure are crucial for the future of Israeli society.

Notwithstanding this fierce opposition, recent years have seen changes in Haredi education, which scholars are only now beginning to describe. There have always been Haredi institutions that have included general education in one form or another. However, recent decades have seen a growth in the number and a diversity in the options. For example, the establishment of the State–Haredi school network in the Ministry of Education (Mamlachti–Haredi). This led to the establishment of schools that offer Haredi families an option for a rich religious curriculum with substantive secular education, while maintain-

ing Haredi cultural norms.[7] Another example would be academic programs targeting the Haredi population that are conducted by Israeli academic institutions in several disciplines, mainly those who tend to be described as vocational, such as law, business management, health-related training, etc.[8] These programs offer Haredim, culturally sensitive programs that lead to academic degrees and which consider the lack of foundation in secular education that many of these students have.[9] Recent data show that, while these programs are growing by leaps and bounds, the overall percentage of Haredi students is still low: the 13,000 Haredi students enrolled in the 2019/2020 academic year comprise only 4% of the entire student population.[10]

These programs allow the graduates to be integrated in the economic life of Israel and find jobs that were, in the past, closed to Haredim. This process, however, is deemed to be slow by many. The gradual decline in government support for families, in addition to the changing norms in the Haredi community surrounding materialism,[11] seem to further the interest in vocational paths; however, at the same time, these changes have attracted fierce criticism from community leaders who see this as a threat to the spirituality of the community and as a pathway for the influence of liberal values and norms.[12] The scholarship on this topic glosses over the internal opposition that faces these academic programs. But it is worthwhile explaining that, at the core of this critique, there are powerful arguments that have a significant effect in the Haredi community. A number of years ago, in a public forum of the online journal *Tzarich Iyun*, the topic of Haredi students in academia was discussed (I took part in this forum and wrote one of the articles). The editors choose to include a short public speech given by a rabbi at a Beis-Yaakov convention in Israel. The speech characterized the dangers of academia as: "A culture war... the secularists and the half religious half secularists do not want us to stop performing *mitzvot*. [they say]: Please, perform *mitzvot* but be like us, live with us, have the same interests, the same values".[13]

The existential experience of the Haredi student is that of loneliness (as another participant in the forum mentioned). Rabbi Nechemya Steinberger, the director of the Haredi Students program at Hebrew University in Jerusalem, surveyed the difficulties faced by typical Haredi students. Some were sociological, and others were—at least potentially—theological. This sense of loneliness, he wrote, can only be dealt with by creating a sense of community that will situate the Haredi student within the Haredi cultural context.[14]

The existence of Haredi programs in Israel is a given at this point; it is also a given that the quantitative data show a clear obstacle in the growth in the participation of Haredi students in both vocational training and the labor market. The internal discourse in the Haredi world seems to have been overlooked during this discussion on a regulatory level in Israel. The latest example would be the supreme court case concerning gender-based programs for Haredi students. The basic question of what Haredi students want is quite unclear.[15]

This internal point of view of the process means that a comparison to the U.S. Haredi community is a very effective tool.

A more complex view in the non-Israeli Haredi world has been of interest among scholars for some time. The comparison to the U.S. is a natural trajectory when it comes to the labor market.[16] This is a question that piqued the interest of Amiram Gonen over twenty years ago, and which brought many other scholars to analyze the ways in which the Haredi community in the U.S. forged a complex coexistence with secular studies and the labor market.

### 3. The Connection between the Haredi Communities in Israel and the Diaspora: Theoretical Framework

What is the nature of the connection and relationship between Haredi communities inside and outside of Israel? On the one hand, there are similarities in external characteristics such as clothing and shared values, but there are also some important differences, such as economic structures. Kimmy Caplan noted that the English-speaking Haredi presence

brings with it changes to the social fabric of the Haredi community in Israel: "Finally, the presence of American Yeshiva students should be seen in the context of the growing numbers of Haredim from English speaking countries especially the U.S. who relocated to Israel. Their presence in the heart of Haredi populated neighborhoods, such as Jerusalem's Meah Shearim and Geulah, has a growing influence on the public sphere, as can be seem on Geula Street".[17]

While an important part of the Haredi discourse in Israel is the rejection of Western culture, the Haredi communities in the U.S. are a conduit to ideas and norms from Western culture. Caplan explains this through it being a result of the historical legacy of American Orthodoxy that was less insular; additionally, it can be explained by the fact that these communities are middle class in nature, allowing for a consumer culture that limits insulation.[18] This last point is very apparent when looking at the most striking example: Kiryas Joel. This village in the U.S. was experiencing great poverty in 2010, redefining poverty in the context of the U.S. By combining both community and Federal and State assistance, even families with limited income could live a life with food security, housing, transportation, and education, without having an experience of poverty. This redefinition of poverty—as both The Washington Post and The New York Times dubbed it—is a great example for the very different experiences of Haredi life in the U.S.; these experiences do not center around poverty and have a consumer culture that resembles the larger American public.[19] The Haredi American model represents a more complex relationship with Western culture: "American Haredim, like middle class American Protestant fundamentalists, seem to understand that one can acculturate without assimilating. However, upholding Torah is paramount, and therefore they tread very carefully between the potential threats of "American" or "Western culture", in order not to lose the bonds that keep followers within the Haredi folds, By living in the West, Haredim have adopted a *modos vivendi* with Western Culture".[20]

It is useful to analyze this relationship from the prism of the theory of transnationalism; this idea focuses on communities of immigrants that maintain their connections with their country of origin and who choose a path of acculturation rather than assimilation in their new country.[21] This is a recent phenomenon; in the past, immigration meant assimilation within a generation. The ease of international communication, travel, and access to the internet [22] allow for a sense of transnationalism; this is a way to acculturate in a host society while maintaining a strong sense of national particularity.[23] The relationship between Haredi communities in Israel and the diaspora has many traits that fall under the title of transnationalism. I believe that the fact that the religious leaders of Orthodox Jewry were almost exclusively centered in Israel after the Holocaust created this relationship that positioned Israeli Orthodoxy as center and the diaspora communities as periphery. The technological innovations of the second half of the twentieth century made the relationship between this center and periphery relatively easy, allowing this model of transnationalism to evolve and be sustained.

Concerning our discussion on the connection between education models in the U.S. and those in Israel, it is worthwhile to utilize a term coined by Thomas Faist: 'transnational social spaces'. He uses this term in opposition to the traditionally discussed 'social fields'. Social fields are generally defined by physical features, boundaries, or territories, and they maximize the role of governmental and intergovernmental organizations as primary actors in transnational exchange. Faist argues that social spaces, on the other hand, transcend the physical world to include communications technology, and are characterized by "the circulation of ideas, symbols, and material culture".[24] These social spaces, more encompassing and inclusive than social fields,[25] are especially valuable to the transmigrants who are constantly negotiating the different, and possibly dueling, identities of their home and host societies. The field of education is an example of such a transnational social space. Physical schools are social fields, where the curriculum and educational agenda are imported from

Haredi educational systems in Israel. This conduit allows for a dialogue concerning both education and other societal values and norms.

This conduit is not a one-way street; it functions in the same way that communities in the diaspora used it to import from Israel—it has the capacity to be a two-way street. In 1990, Amnon Levi had already pointed to the possible influence of the U.S. Haredi communities in Israel;[26] Kimmy Caplan's work showed that, indeed, this is an evolving process that suggests several paths of influence.

## 4. Monsey Jewish Community: Background

The Jewish community in Rockland County has been growing exponentially in the past decades. The soaring prices of housing in Brooklyn have brought Jewish Orthodox families to look for affordable housing options in the Monsey area of Rockland County, NY, and the Lakewood area in Ocean County in NJ. The families moving to these destinations are young and the high birth rate in these communities creates a vibrant and constantly expanding educational network.

The goal of this paper is to examine the framework of existing secondary educational institutions in Monsey, specifically the scope of secular studies and trends in the community concerning higher education. Based on the assumption that education is a tool that enables social and economic mobility, the paper sets out to examine what actions can be taken to increase and diversify educational opportunities in the community. The data and analysis in this paper can be used as a foundation for discussion concerning the social advancement of the Monsey community and for identifying similar strategies that can be used in Israel. There are several key questions that need to be answered when it comes to policy: Is the goal to lift people from poverty and enable them to be independent from social security assistance? Is the goal to integrate members of the Haredi community into the U.S. middle class? Is the goal to encourage people to acculturate?

This complex and wide array of educational institutions deserves special attention for several reasons:

1. It is a network of institutions that represent almost the entire gamut of educational perspectives and cultures within Orthodoxy (barring co-educational modern Orthodox institutions) and could serve as a test case for the future of Orthodoxy in America.

2. Since Monsey is not too far from the community centers in Brooklyn, it has the ability to affect policies and cultural norms in these communities. At the same time, the fact that Monsey and its educational institutions are not firmly in the center of these communities and their leaderships creates the potential to be open to educational experimentation and initiatives.

The community has been showing consistent growth for many years. The 2010 federal census found 311,687 residents in the county and estimated that in 2017 there would be 328,868 residents.[27] In the census, there was no category for Orthodox affiliation and it is most likely that Jewish–Orthodox families identified as white. Based on that, it seems that 30% of the population are younger than nineteen;[28] in the county, 7166 families identifying as white reported receiving nutritional assistance in the form of food stamps.[29] Only 40% of the population in the county had a bachelor's degree,[30] and in 2010, 6.68% reported that Yiddish is their first language and 1.15% reported that Hebrew is their first language.[31]

The majority of the children in the county are part of the East Ramapo school district, but most of the children in the district attend private educational institutions. In the school district, 8900 students are attendees at 14 public schools, compared to 20,000 students attending 80 non-public schools, according to the East Ramapo board of education.[32] The changing demographics and the limited financial support from the school district to the non-public schools has been a constant political flashpoint between the Orthodox community and the larger population. As time went by, the majority of the board members were Orthodox men, and they were accused of being responsible for the deterioration of the public school system. Menachem Berkowitz, a member of the school board, estimated

that these numbers are likely not accurate in light of the constant growth in the number of students in private institutions and the limited information the board has on private institutions; primarily, he cited the fact that private schools that are not receiving any services from the board of education will not be on record.[33] Berkowitz noted that, after years of tensions between the Orthodox and the non-Jewish population concerning the school board, recent years saw a positive atmosphere and growing collaboration. The private schools receive support in the form of busing, special education, psychological services, and financial support that is given per student for security; additionally, textbooks and other services are provided which are not connected to the religious goals of the schools. While the estimation concerning a more positive atmosphere might be contested, it is clear that the only population growth in the district is within the Orthodox community, making the political strife concerning the school district and its finances less volatile.

Politically, the Jewish community in Monsey has a very limited view of the political process. Members of the community are involved in issues that directly relate to them, such as funding for busing and zoning for synagogues and schools. However, there is hardly any public discussion on issues concerning larger political questions. For example, in Yiddish newspapers, voting drives exclusively target the Democratic party as it has been determined that the community can gain the most by voting for the Democratic party. In another example, New Square, a small village populated exclusively by a Skver Chasidim community, votes as a bloc for the Democratic party because of a longstanding alliance with the Clintons, following Bill Clinton's pardon of a community member during his presidency. Haredi isolationism prevents it from having a meaningful part to play in the American political process.[34] The Trump election in 2018 created a shift in the political identification of the Haredi community, but there remains a difference between the presidential elections and the local ones as far as voting patterns are concerned.[35]

The Orthodox community is not homogenous; it divides into many factions and sub-factions that differ from one another in many ways, some big and others small. As a result, there are different educational models in Orthodox communities that need to be discussed separately in light of differing views on secular education and engagement with American society and the job market. This report will begin with a general survey of the different communities in the Monsey area and will later continue with three separate chapters discussing the major educational categories prevalent in Monsey: Litvish institutions, Chabad Yeshivas, and Chasidic Yeshivas.

## 5. The Social Framework of Monsey

There are many differences between the many communities in Monsey, but there is a wide common denominator that overlaps the entire gamut of the Haredi world. First and foremost, there is a basic ethos of a rejection of modernity. This creates a sense of insularity and a very suspicious attitude towards norms coming from the non-Jewish world. Another important pillar is the centrality of Torah study; the study of the Torah is presented as the goal every Jew should aspire to—leaving everything else as secondary. Part of this notion is that, for the sake of Torah study, physical comforts should be sacrificed; therefore, materialism is presented as negative and counterproductive as far as spirituality is concerned.[36]

The characteristics of the Litvish/Mitnagdish community can be divided to several categories. Externally, most of the men in the community will not have beards unless they hold a religious position of some significance; the men will wear modern suits and will speak English for the most part. The community emphasizes Torah study as the center of religious life and celebrates a model of a man that will dedicate his life to Torah study at the expense of economic success. This Orthodox community in Monsey began its growth in the 1950s, when a prominent Litvish Yeshiva, "Beis Midrash Elyon", opened its doors in Monsey and attracted families from Brooklyn. As the years progressed, Lakewood became the center of the Litvish community in the U.S., thanks to the establishment of the world's biggest Yeshiva by Rabbi Aharon Kotler in Lakewood, NJ. That being said, in light of

the housing costs in Lakewood, there is continued growth in the Litvish community in Monsey—though it is secondary to the rise in the Chasidish population.[37]

In Monsey, the Litvish community is decentralized, and there is no dominant rabbinic leadership that spans the many Litvish synagogues in the area. This community is best described as a spectrum rather than a binary. There are members of the community which are integrated into secular society, holding professional jobs and supporting educational models that integrate secular and religious studies. Others are more critical of the integration of secular studies and might promote an ideal model of a Kollel student who learns all day and receives community support by those members who cannot or will not commit to a life of study; therefore, these individuals peruse a professional career and fulfil their religious duties by supporting those who are interested in a full-time commitment to Torah study. In other words, the Litvish community as a spectrum has a side to it which finds much in common with the right-wing modern Orthodox community as far as financial expectations are concerned.[38]

Another distinct community in Monsey is Chabad–Lubavitch. This Chasidic group includes several hundreds of families but is growing every year, because Monsey is still an affordable option for families who cannot afford housing in Crown Heights. This community has an outreach mentality that is evident in the thousands of outreach centers created by this community all over the world. This being said, not all of the members of the community are comfortable with interactions with society at large; those who do not go to establish outreach centers create communities that exist with an inherent tension between insularity and interaction vis à vis secular society.

The ethos of this community centers around a robust tradition of intellectual religious experience, specifically of theology (differentiating it from the Litvish community, where the intellectual pursuits are firmly situated on Talmudic texts). In the past sixty years, this ethos developed to include the teachings of the last Lubavitch Rebbe, Rabbi Menchem Mendel Schneerson, who built his community on the duty to be involved in outreach activities in every possible venue. Such an agenda brought about the participation of members who are committed to outreach work, as well as that of community members that are less insular, in comparison with other Chasidic groups. This results in a willingness to venture outside of the Jewish community when it comes to business or the job market.[39] The importance of the Monsey community for Lubavitch is twofold: First, it is close to the heart of the community in Brooklyn, which is a main attraction that brings a constant flow of families to move to the area; the area is more affordable than any other alternative in the NY metro area. Second, it is far enough from Brooklyn to be comfortable to test community boundaries and norms since it is not in the immediate vicinity of leaders in the community, but it is still close enough to be visible to the Crown Heights community and cause residents of Crown Heights to rethink and reevaluate the way they do things.

The Chasidic community in Monsey is very large and consists of several sub-groups. The communal lives of these communities are conducted exclusively in Yiddish; in the cases of Square, Vishnitz, and Satmar, it is easy to find men and women who are third-generation American citizens who speak broken English. This communities which are mostly of Hungarian–Romanian decent are more extreme in their rejection of outside influences. There are other smaller Hasidic communities in the Monsey area which are less restrictive in light of their smaller size and the fact that their community centers are located in Israel or Brooklyn. This is the case with Ger, Belz, Tzans, Bobov, Karlin, etc.[40] These are Polish Hasidic groups and are slightly less insular and less critical of the state of Israel.

## 6. Chasidic Educational Institutions

In the Chasidic community, there is a clear hierarchical structure. At the top of the pyramid is the Tzadik, usually referred to as a Rebbe or Admor. In the established and large communities, the Rebbe or Admor has the last say on community policy and life and sometimes on the personal choices of individuals. In Monsey, the largest communities—

namely Skver and Vishnitz—have their Rebbe living in their immediate vicinity, making his control more felt and present.

The governing ethos of the different Chasidic communities is insularity. These communities pride themselves on building walls between themselves and the outside world; this is exemplified by their unique clothing, language, and their public rejection of traditional mass media and social media. However, in the past decade, there have been changes in the levels of insularity. The internet and the ability to access it with anonymity—while hiding this access even from members of the family—has been creating shock waves in Chasidic communities. This new reality has had significant impacts on the community in many different ways—economically, educationally, and normatively.

There are many members of the community who are opening businesses which function in an online environment; e-commerce allows members of the community who have limited secular education to open a business and go beyond the boundaries of the community. This leads to a greater level of involvement with the outside world and its norms and values, leading some to explore ideas and possibilities outside Monsey. This has also brought about negative outcomes, namely an increase in drug use leading to a growing rate of deaths due to overdosing.[41]

Within Chasidic communities, there are attempts to combat the growing influences of modernity, mainly by attacking the main vehicle: the internet. There are continuous and extensive campaigns warning against the use of the internet and smartphones. Newspapers, magazines, and mass gatherings are continuously voicing the "dangers of internet"; the use of the internet can lead to sanctions such as rejecting children from admission to schools. The fear of these changes is based in reality; there is a growing constituency of families dubbed "modern Chasidim". This term refers to Chasidim members who are not comfortable with the yoke of insularity and choose to change their lifestyle from the community norm. Some will introduce English books to their homes, use the internet, change their attire to a degree, and change the schools their children attend to more liberal and less insular institutions.[42]

These communities' characteristics are extremely important as far as the discourse and action concerning social changes are concerned. The different sub-groups of the Monsey Orthodox community, while having much in common, have different levels of tolerance to modernity and have different needs as far as their educational institutions and economic structures are concerned. Any kind of project in Monsey attempting to bring about social mobility and broaden the economic venues that are open to members of the community will need to be tailored to these cultural and normative boundaries. Without acknowledging these sensitivities, any such project is doomed to fail.

## 7. Litvish Secondary Education

The secondary education institutions for Litvish boys can be roughly divided to two categories: Yeshivas with secular studies that fulfil the minimum requirements of the state of NY as far as secular education is concerned, including the regents exams; Yeshivas which are exclusively centered on Torah study and include no secular education at all. Most of these institutions do not have boarding options, so the students are local for the most part; this makes these institutions smaller and very connected to the culture of the local community.

The daily schedule of the students is demanding. The day begins at 7:30 a.m. with davening; after breakfast, Gemara studies begin at 9:30, which will include independent study and a Shiur (a class); lunch is at 1:00; this is followed by another seder (section of the day dedicated to religious studies) at 2 p.m. In Yeshivas with a high school component, secular studies will take place between 3:30 and 6. In almost all the Yeshivas, students will go home at 6 only to return at 7:30 or 8 for "night seder"—another period of an hour or ninety minutes for Talmud study.

Traditionally, in the United States, the Litvish Yeshiva for high school age, usually referred to as *Yeshiva ketana*, includes secular education. Institutions such as Torah V'Daas,

Chaim Berlin, Telz, etc., were created in America and integrated secular studies with the understanding that their graduates require a pathway to livelihood. However, this model is facing competition from what has become known as the Lakewood Model. In Lakewood, the biggest Yeshiva in the world, Rabbi Aaron Kutler created an ethos which was known in Yiddish as *nar Teyreh* (only Torah). This ethos, which until this day controls the Haredi discourse in Israel, proclaims that children and young men should study only the Torah, disregarding issues of potential employment in the future.[43]

In America, these two competing models live side by side. In Israel, there is severe criticism and vitriol against the inclusion of secular studies in Haredi institutions; however, the American Litvish community is able to take a different approach while still maintaining its veneration of the leadership of the Haredi world in Israel. This dissonance is practical, not ideological—there is no core ideological explanation for this difference; it is simply a fact of life that, in America, things are carried out differently. That being said, there is considerable room for improvement, in light of the fact that some institutions reject secular studies; even those that do include secular studies send a conflicted message concerning secular studies. Namely, secular studies are secondary and, at best, a tool to generate income; there is a belief that even true men of faith will not require this knowledge to earn a living and God will provide for them if they truly commit to Torah study.[44]

In an interview I conducted with a longtime educator in a large Litvish Yeshiva that includes secular studies, he states that: "Beis-Shraga is a great example for a Litvish *mosad* (institution), the goal is to take a soft *yeshivish* home and make them strong Litvish. They offer regents curriculum but would prefer the students don't go to college, but they are not very worried if some do".[45] The preferable, "safe" option is for graduates who do not intend to continue to learn and to teach Torah to turn to commerce or another vocational path. Clearly, there is a connection between success in the job market and college degrees. The college option is problematic in the Litvish world, since college is identified as a center for promiscuity on the one hand and atheism on the other.[46]

Obviously, such an attitude creates a significant barrier between graduates of these Yeshivas and a typical American college. The college campus is considered foreign in their eyes. This approach has impacted the attitudes among the students; when asked what the boys think of secular education, one student answered as follows: "Boys are interested in a diploma, some are naturally curious. But there are no AP's or CLEPs; all the teachers but one is Jewish and religious, they do not see people from the outside".[47] There are also objective barriers between the community and colleges, namely the fact that male students finish their studies in the Yeshiva system between the ages of 22 and 25. The second barrier is the price tag of most American colleges, which is prohibitive for many families in this community.

Unlike the *Yeshiva ktana* high schools for boys, the secondary education institutions for girls follow a different path. The Litvish high schools for girls, usually referred to as Bais Yaakov, tend to be more inclusive as far as the communities that send their daughters to them. The Bais Yaakov institutions have a range of attendees, from girls form more modern Litvish families in some Bais Yaakovs to Chasidish girls in the more traditional institutions. In an interview I conducted with Rabbi T., an associate principal at a Bais Yaakov in Monsey, he noted that the educational visions for girls can be divided into two categories. The first is a pedagogical agenda that wants the graduate to be a mother and homemaker. This message can be found in institutions, such as Bais Rochel, where secular studies hardly exist, and an entrepreneurial graduate will most likely open a gift shop or another other small business to express her creative abilities. But Rabbi T. continued to say that some institutions understand that, in today's economy, a graduate needs to have career options; they understand that the graduate should have the tools to acquire jobs that will include some kind of training outside the Jewish world.[48]

In the past fifteen years, Litvish institutions have created certain pathways to higher education. These pathways have recently also begun targeting male students of Litvish Yeshivas, which brings us to the attitude of these institutions to higher education. In

Lakewood, there is a tradition of criticism of higher education, and graduates of Lakewood, as I mentioned before, criticize any attempt to acquire academic credentials. However, ever since the 1950s, other Litvish Yeshivas were tolerant of their graduates attaining some kind of academic training in non-Jewish institutions. Currently, there are several avenues open for male or female Litvish graduates who wish to attain academic credentials to further their career in the job market. These graduates will return to the U.S. after spending time in Israel for religious studies. The average male student will be between 22 and 25 years old, sometimes older. The female students will be 20–21. Upon their return, they might integrate studying in a local academic institution in the afternoon, while spending their mornings in a religious environment such as Yeshivas or post-seminary programs. There are still Yeshivas in Baltimore and Brooklyn where you can find students studying the Torah until 3 p.m., when they then begin their academic studies at local universities and colleges.[49]

Another avenue is dominated by certain private and public colleges and universities who identified the potential of the Haredi market. These universities and colleges include institutions such as Concordia College, Bellevue University, Thomas Edison State College, Empire State College, and Fairly Dickenson University. They have developed avenues for a BA for Haredi students. These institutions offer Haredi students a shortened period of study by accepting credits from Yeshivas and seminaries in Israel with A.P./CLEP credit. They are willing to accept transfer credits, sometimes up to 90 credits, leaving a student with only one year of study to complete in order to be granted a BA These programs offer a combination of online courses and traditional courses held in Brooklyn, Monsey, and Teaneck, which are open only to Haredi students and are gender-segregated.[50] These pathways will grant students a BA in business management and, in some cases, education. Other programs offer a degree in accounting. There was an attempt by Touro College to create a more rigorous BA program for the Haredi world, under the title Machon L'Parnasa (Institute for Livelihood), but in the years since its inception it has not gained popularity. Several interviewees explained that its small size results from the fact that it is considered a second-rate institution that is run by Jews and targets the Jewish community; additionally, and perhaps more importantly, it requires students to go through the entire process of academic studies to achieve a BA, which many students feel is unrealistic due to their age.[51]

In an interview with a director of a small program granting bachelor's degrees, the cultural context of the programs was emphasized. The programs chosen by most of the Haredi students are those that accommodate their cultural sensitivities and first and foremost take place in a "yeshivish setting". These programs either take place online where gender boundaries are not needed or, if they do take place in a traditional classroom, they are single-gender-based.[52] This seems to be an essential part of the integration of Haredim in higher education.

The most significant challenge, as it relates to the Litvish community, seems to be divided into two tiers. The first is the fact that there is a certain constituency within this community that sees secular studies as spiritually dangerous, and which is not creating paths for its young people to enter the modern job market. This reality, which originates from the Lakewood community, became such a strong ethos of the Lakewood culture that it overpowers even the current leadership of the Lakewood Yeshiva. In interviews I conducted with community activists in Lakewood who enjoy a good working relationship with key figures in the Lakewood Yeshiva, they indicated that the leadership understands that the current economic structure is not viable. However, they seem to lack the ability to publicly state the need for a change.

The second tier has to do with the current higher education options that are available to those members of the community who do not follow the Lakewood ethos. As I previously indicated, the colleges and universities that work with the Litvish community created specific tracks that tend to be focused on business, education, and accounting. The main difficulty is that these degrees do not have the same social capital they might have had in previous years. The changing nature of the job market requires the community to open and access new venues of higher education, specifically those that can be easily translated to

the job market: for example, computer science, technology and engineering, nursing and assisting physicians.

## 8. Chabad–Lubavitch Yeshivas and High School

The Chabad–Lubavitch community is different from other communities on the spectrum of the Haredi world. It is more integrated in the non-religious and even non-Jewish world at large, and is not invested in creating an enclave society. Its theology provides underpinnings for engaging with the world in order to foster the world's moral and spiritual growth and to actualize the spiritual potential of worldly objects and projects. There is a self-confidence born of a long history of strong organizational and intellectual abilities. The constant influx of *Ba'alei Tshuvah* (people who opted to become religious) also brings with it more points of view and approaches to education and different needs as far as curriculum is concerned.[53]

The Yeshiva for high school boys in Chabad communities is referred to as a *Mesivta*; you can find them all over the world, and they are usually small institutions with populations ranging from 40 to 150 students. The daily schedule in such an institution is very similar to the Litvish *Yeshiva ktana*, with one major exception: a third of the day is dedicated not to Talmud study but the study of Chasidic texts, written by the leaders of the Chabad movement. The vast majority of Chabad Yeshivas did not include a secular curriculum; there are exceptions in the Yeshiva in Toronto, the Ocean Parkway Yeshiva in Flatbush, and Yeshivas in Pittsburg, Australia, and a few more locations.

What is the catalyst to the inclusion of secular studies in Chabad? It seems that this phenomenon can be traced back to several factors. First, the fact that Chabad communities in some cases create outreach institutions. Chabad schools target populations which are interested in Jewish education, but which demand secular studies that are on par with the educational norms in their state. This fact brings some communities to open such schools and send children from Chabad families to these schools alongside children from non-Chabad families. Such developments in smaller Chabad communities have a cumulative effect on the larger community and bring about calls for changes in education. A second cause is the changing economic vision of the central hub of Chabad in the Crown Heights community. In recent decades, many young couples saw their future as *Shluchim* emissaries of the movement, establishing outreach centers in various places around the globe. In the past decade, it has become apparent that there are few new options for such new outposts. Young families in the community are turning to other career avenues in business, commerce, etc. The growing number of young men and women in the community who are searching for career paths with varying levels of success brings parents to discuss the option of adding secular education to secondary education institutions.

Most girls' institutions in Chabad include secular studies; in NY, for example, the Chabad high schools for girls are based on the curriculum created by the board of regents of the state of NY. The official goal of these schools is that their graduates obtain a NY regents diploma—a diploma that opens paths to further higher education. The differentiation between male and female education shows that the opposition to secular education in the *metivta's* has more to do with sociology than ideology. In an interview with a businessman from Monsey who supports one of the Yeshivas in Monsey, he said the following about Yeshivas that include secular studies: "in the Haredi Chabad world non-religious activities integrated in the curriculum are seen as catering to kids at risk... kids on the periphery of religious observance. They have challenges with their observance".[54] When asked about the fact that there are Chabad institutions that do offer secular education and are catering to the core of the community, he answered: "To a degree they are considered outside the box, but the Monsey Yeshiva is not main stream but we are trying to be that, we are trying to inject certain experiences to Jewish education. I believe in what the Gemara says יש חוכמה בגוים אין תורה בגוים (there is wisdom among the gentiles there is no Torah among the gentiles)".[55]

In Monsey, there were two secondary education institutions: one for boys and one for girls. The boys Yeshiva has gone through many stages and changes in recent years. It operates with anywhere between 50 and 70 students in rented facilities. This Yeshiva offers no secular studies, and it could be construed as a "typical" Chabad Yeshiva. The students come from Monsey for the most part, while many come from out of town, including Brooklyn. The boys are expected at the conclusion of their three years in the Yeshiva to continue to a *Yeshiva gedola*, the next phase in their religious education. There was a second institution that was created in 2015, and which closed after the COVID-19 outbreak, under the title *Lev Temimim*.[56] This Yeshiva was geared toward what the founders perceived as the vast majority of teenage boys in Chabad communities. In other words, not excellent students who can study Torah all day long and excel in it, but average boys with average talents. The Yeshiva was founded with the goal of giving these boys an opportunity for religious growth and self-value even without the ability to excel in religious studies. The Yeshiva integrated secular studies for various reasons. In interviews with several stakeholders in this Yeshiva, it seems that the integration of secular studies defined the Yeshiva as a Yeshiva for weak boys. It is interesting to compare this perception of *Lev Temimim* to the local high school for girls. The latter, called *Bais Chaya Mushka*, was founded and is led by a Rabbi who holds a doctorate in physics and operates a full and successful secular studies program that qualifies the girls to take the New York Regent exams. Girls take AP exams in some cases, and the secular studies program is a natural and valued part of the curriculum. This striking difference proves that the opposition to secular studies within Chabad Yeshivas has much to do with sociological perceptions, and when those change, many members of the community would appreciate this option.

What are the core ideological and sociological obstacles to promoting such a change? In Chabad, there are many Chasidim that will claim that the Lubavitcher Rebbe objected to secular studies. While the Lubavitcher Rebbe did object to secular studies on some occasions, "When the Rebbe said no college it was 1960s' it was said in a very specific context namely colleges in the 1960s' and their campus culture".[57] It is intellectually dishonest not to contextualize this statement in this day and age. In reality, the Lubavitcher Rebbe had a much more nuanced, complex point of view concerning secular studies. Well beyond the fact that he himself was educated in academic institutions in Berlin and Paris, he addressed the issue in 1978 in one of his theological discourses; here, he made it clear that there is religious value in approaching secular studies and revealing Godliness within them.[58]

The dissonance between the economic needs of the community and the perception that there is a spiritual problem with secular studies can be solved in the context of Chabad by rebranding secular studies. In an interview with a Rabbi of a Chabad synagogue in the Monsey area, he notes that the young adults in the community—usually referred to as *yunge leit*, in Yiddish—have pursued and are pursuing training in non-Jewish institutions post-marriage. He gave several popular examples: accounting, physician assistance studies, nursing, and coding. The popularity of these tracts suggests that, in his mind, integration of studies branded as professional training within the educational system will be perceived very differently than merely the addition of secular studies to the curriculum. Finally, it is worthwhile remembering that a significant catalyst for changes in education has to do with tuition. The price tag associated with religious education is high, and a scenario that will allow a Yeshiva that offers secular education to cut tuition will enable it to be much more selective and to significantly increase its pool of applicants well beyond the boundaries of Monsey.

The vast majority of young men in Lubavitch finish their Yeshiva education at 23–24; young women will be finished by the age of 19–20. Many of these young adults will marry soon afterwards, with children often coming quickly. The ability to begin academic or vocational training is obviously inhibited by the realities of starting a family. Similarly to Litvish couples, the need of young adults in Chabad is for academic or vocational training that will quickly lead to gainful employment. Surprisingly, most observers point to the fact

that young Chabad men and women do not turn to the specialized short pathways to a BA when compared to their peers in the Litvish world. It is hard to explain why this is the case. Perhaps this has something to do with the culture in Chabad communities, which is very practical and to the point; this leads young men and women to start their own businesses or to prefer more vocational training. In 2017, it seems that forces within the Crown Heights community were able to convince Medgar Evers College, in Brooklyn, to accept Yeshiva credits; they pointed to the fact that there are forces in the community that believe relationships with colleges should also be developed, similar to those that exist in the Litvish world.[59] There are many forces within the community that cultivate this culture of entrepreneurship, such as Crown Heights Young Entrepreneurs, an organization geared toward helping *yunge leit* open their own businesses. It seems that programs that will be branded as helping to foster this culture will be accepted as legitimate in the community. This would be an opportunity to introduce academic studies. In the long run, it could also legitimize interest in other fields of knowledge and study beyond the more popular fields of science, technology, accounting, and others, as mentioned above. A homogenous cultural learning environment will be beneficial in this case also, like that in the Litvish community.

## 9. Chasidish Secondary Education

As I mentioned in the opening of this paper, the Chasidish community is governed by an ethos of insularity. This is exemplified by the clothing, language, official criticism of mass media and social media, and other factors. But this ethos is the strongest in the education system. Inculcated in the students from a young age is the idea that the outside world is a combination of danger and spiritual inferiority. Part of this ethos is the rejection of secular studies. Today, in Monsey, there is hardly any kind of secular education in Chasidish institutions.

This has not always been the case. In the past, even the most stringent and insular communities included secular studies in their elementary education. Today, boys' institutions have the bare minimum of secular subjects—if any at all. Obviously, no secular education exists in secondary education for boys: "The hard core ones have nothing past 12 years old and there is only 45 min a day. The rest: Belz, Ger, Vien have nothing past 14. It used to be different in Bobov, Vien perhaps Ger had regents exams. Even Satmer had 3 h until the age of 13. I don't think they will change their minds if they will be forced from the outside".[60]

The above quote is very revealing. It seems that the gradual digression from secular studies in the Chasidish world had to do with a sense of religiosity. One institution started to diminish the amount of secular studies, and others followed suit, fearing that they would be perceived as less religious. Even though the community often seems to be a homogeneous group, in reality, under the surface, there are a range of opinions regarding secular education. Karlin—a Chasidic group with a medium-sized following in Monsey—is led by a Rebbe who is American-born and -raised, and is now living in Israel. He is known to have an open-minded and sophisticated world view; the community in Monsey seems to have many forces within in that would be interested in developing more educational and economic possibilities for the younger generation. On the other hand, in the core of the Viznitz and Satmar communities in Monsey, you will find young men who are third-generation American citizens but speak broken English, to whom English is a second language.

In an interview with an educator from the Chasidic community, he suggested that there could be—perhaps since there are a few scarce examples of secular education in Chasidic institutions—room to establish a Chasidic Yeshiva that includes secular education. The reasoning behind such a suggestion is the objections behind secular education have more to do with sociology than anything else. In other words, the establishment of an institution that will maintain all Chasidic characteristics (attire, mikveh, daily schedule, etc.) and which will also include secular studies will be perceived as legitimate by many in the Chasidish community. While they will not be relevant to very insular communities such

as Satmar and Skver—where students have almost no exposure to secular studies—and will need considerable remedial work, this might be an option for others that come from less insular communities and families. In an interview with another community activist, he suggested that, while this might be a very ambitious project, existing institutions might be willing to integrate programs that will be seen as orientated toward vocational training, basic programming, accounting, etc.

Like the other communities surveyed before, there is a difference between secondary education for males and for females. There are more educational possibilities for Chasidish girls, including—in some cases—Regent exams. This is the case with Bais Rochel, a highly traditional school that caters mainly to girls from Chasidish families. You can also find Chasidish women using online programs to obtain degrees, but it is difficult to ascertain the scope of the phenomenon.

For young men in the community, who usually start families at a very young age, there are several trajectories when it comes to providing for their families. Many start businesses which are connected to commerce and e-commerce. There is also a willingness to obtain blue-collar jobs. During an interview with a local businessman and community leader, Rabbi Yaakov Horowitz, he suggested that there is room for the establishment of a trade school. He believes there will be significant demand in the Chasidish community for this kind of training, including—but not limited to—air conditioning technicians, refrigerator technicians, mechanics, carpenters, etc. He mentioned that, in the past, he attempted to cooperate with a state-sponsored institution training men and women to enter this market. However, there are some cultural challenges that make it difficult to integrate Chasidim from Monsey directly into classrooms in such institutions. Some of these difficulties are technical—mainly the fact that they have a significant part of their classes on the weekends and will not accommodate Jewish holidays.

However, there are also more fundamental challenges, mainly those which arise when it comes to the integration of Chasidic men and women who come from such an insular environment with students from completely different cultural backgrounds. Its goal is to allow young men from the Chasidic community there to integrate into the job market and it has operated successfully for the past several years. In an interview with community activists in Lakewood, they mentioned that many Chasidim in Lakewood work in such positions. Women in this community will also be interested in vocational training, especially in the fields of accounting and coding.

The unique cultural characteristics of the Chasidic community require a nuanced and culturally sensitive approach in order to strengthen the ability of young men and women from this community to take a larger part in the modern job market. While sensitivity is required in relation to the communities discussed before, the Chasidic community has more cultural barriers, such as language, fashion, and suspicion of outsiders. That being said, there are many forces in the Chasidic community that understand the urgent need to implement changes in the education and training system that is available to community members; this is an excellent opportunity to advance, and to make such important changes.

In the case of the Chasidish community, it is hard to see an easy path to civic engagement. The level of insularity and the high rate of poverty in the community, accompanied by a high level of dependency on social security, creates significant barriers towards civic engagement. I do believe, however, that the creation of vocational training programs and attempts to improve secular studies in secondary education will result, in the long run, not only in a new economic structure for the community but also in a new political structure.

## 10. Summary and Recommendations

The data collected for this paper suggest that there are several directions that can be pursued in the attempt to increase the educational opportunities that are available in the Haredi communities in the U.S. and in Israel. These need to be sensitive to community norms, which differ considerably both within the Monsey area and in Israel. The benefit of using the model developed in Monsey is the fact that, in Monsey, this process is in a more

advanced stage, albeit far from finished. The changes in Israel began a little over a decade ago and are still in their nascent stages; meanwhile, in the U.S., the integration occurring in the Litvish communities and—to a lesser extent—in Chabad–Lubavitch communities began long before that.

The first question is the following: how are the models described here able to circumvent barriers that are so significant in the context of the Haredi community?

The difficulties of implementing changes in the field of education in the Haredi community have been discussed by many scholars;[61] the literature identifies obstacles that can be divided into four categories:

1.  An ideological barrier—the narrative that innovations are prohibited by the Torah is so common in Haredi discourse is a significant obstacle. The narrative used in Monsey is not burdened with this barrier, since the discourse is centered around vocational training or *Parnasa* rather than a liberal education narrative that is based on a specific ideological vantage point.

2.  Organizational Conservativism—in Israel, Haredi education is built around centralized systems such Hachinuch Ha'atzmai or community-based systems. This reality creates a level of resistance to change that is no different to that in any other large institution. This comprises a significant difference between Monsey and U.S. Haredi cultures in general, where the private educational institutions are decentralized to an extreme and there is no concept of chains of schools with consolidated management. This allows for new schools to be established as grassroots action. For this model to be transplanted in Israel, it will also need to be situated outside the existing systems.

3.  Structural difficulties—a point that is reiterated by many scholars is that there is difficulty in overseeing Haredi education and assuring the implementation of changes. In the context of this model, this problem is circumvented, since the market and—to a lesser degree—the regulators deal with the proficiency of the graduates. This point requires further clarification: first, when it comes to vocational programs such as IT, the skill level of the graduate determines the marketability; the same is true for accounting, where a graduate of a program targeting Haredi students will be judged according to their skill level. When it comes to programs like nursing, the licensing authorities will regulate the studies. In other words, the more vocationally oriented the program, the less there is a need to worry about supervision. This approach can be relevant to secondary education, where outside testing makes the final judgment of the quality and efficacy of the program.

4.  Political–economic power structures—in Israel, there is a symbiosis between the educational institutions, many of whom are part of a centralized chain, and the political and religious leadership of the community. This is not the case in Monsey, where the separation of church and state minimizes the scope of the funding that religious schools can obtain from state agencies; therefore, the power of the political leadership is limited. This is an issue that seem to be the most significant barrier to circumvent in transplanting the Monsey model in Israel. The solution—I believe—is mainly to be found in building trust with existing systems through agents of change. A discourse that focuses on *parnasa* rather than on an ideological discourse that will fill the void of vocational training and facilitate Haredi participation in the job market can be accepted as a genuine attempt to improve the lives of community members.

### 10.1. Secondary Education

As noted in the previous pages, there are different models of secondary education in Monsey. Some institutions integrate secular studies, and it seems that many Litvish and Chabad institutions that do not integrate seem to be willing to consider such integration in exchange for financial support. Such a grant needs to be conditioned on objective criterion that will ensure that the money is spent in the desired manner. Since Monsey is in the State of New York, such grants can be conditioned on a certain percentage of the student body passing the Regent exams. A similar model can also be used in Israel where high

school matriculation exams exist and are well-established, supporting a wide variety of learners in different levels of academic skills and rigor. Such a condition connected to a measurable criterion would require that an institution commit significant resources to build up a viable secular studies program. Concerning the Chasidish world, there could be room for the establishment of a Chasidish Yeshiva that would integrate secular studies. Such a project would require a significant grant, but there are local educators who are interested and would be willing to take part in such a project; such a project might have effects well beyond the boundaries of Monsey. In Israel, such a program under the title of *HaMidrasha HaChasidit*, located in Beitar, has been operating for several years.[62] It could very well be that a similar project in the United States will have positive effects. Finally, even if such an ambitious project might be too challenging, there is room to offer grants for Chasidic Yeshivas that will integrate vocational training as part of their secular education. While such programs will need to be more modest in their scope, it will have the potential to cause a shift in public opinion and motivate young men and women to explore new career paths.

*10.2. Higher Education*

As noted, there are certain academic avenues created by Jewish and non-Jewish colleges and Universities in recent years. There is no need to explain that a BA opens doors in the modern job market. That said, there is great room for improvement concerning the existing options. First, many members of Litvish and Chabad communities do not utilize these existing options. Second, the existing programs tend to be oriented toward fields which are not as relevant in the fast-changing job market. There is significant potential for projects that will open up new avenues of academic studies in the Haredi community. Parallel projects in Israel are growing constantly; a centralized effort to work with and encourage local institutions to create such special programs is essential. A focus should be put on the creation of academic programs that are relevant to the contemporary and future job markets. As I mentioned before, it is obvious that many members of the community are interested in programs to qualify them as physician assistants and computer programmers, and in other STEM programs. It stands to reason that this will also motivate those interested in other academic programs to pursue such interests.

In the Israeli context, there has been considerable discussion on the single-gender structure of the Haredi programs. This has been challenged in court and the basic question of the need for such segregation among Haredi students has been challenged by opponents of the Haredi programs in Israel. The examples from Monsey unequivocally show that such separation is a given, and it is difficult to imagine any other model in this cultural context. The heavy use of online courses in American higher education might also help ease some of the tensions as far as hiring professors is concerned, since it seems there is no objection to female professors in online courses for men. If the Monsey case can teach us anything about higher education, it seems that single-gender classrooms are a given for the successful integration of Charedi students in higher education at the undergraduate level.

*10.3. Vocational Training*

Specifically, in the Chasidish world, there is legitimacy and motivation to pursue blue-collar professions. It seems to be the epitome of the American dream—improving one's economic reality through hard work. There seems to be significant interest in the community in the formation of such a program. It is hard to overstate the significance of the creation of many small businesses in a community such as Monsey. It has the potential to create not only economic change, but social transformation and a shift away from heavy reliance on social security and social benefits.

Education can bring about social mobility and change political and ideological positions. Many Orthodox leaders recognize that changes are needed in the Orthodox education system in light of the ever-changing economy. Such changes can and should bring about new ideas on the contribution of this community to larger society, be it in the U.S. or Israel. The Monsey community can be an epicenter of change for the Haredi community in the

United States, and the models utilized in the Monsey context should inform the Israeli process. Concentration on education reform and the introduction of higher-quality secular studies programs in secondary education, as well as more pathways to higher education, will bring about civil discourse in the community and a higher rate of participation in civic life.

### 10.4. Point of Difference

The theoretical model of transnationalism allows for cross-pollination between the center and the periphery; in the case of Monsey, I did not find evidence for the Israeli Haredi discourse concerning secular studies, but that does not mean that this is true for all American Haredi communities. In Lakewood, where the Litvish dominance is much clearer, I found a more substantial rhetoric against secular studies in Litvish circles. This goes beyond the scope of this article and requires a separate research project.

It is also important to note that there is one factor that has a dramatic effect on the comparison between the communities: military service. This issue has—for decades—been a central point of contention between the Haredi community and the Israeli establishment. Opening avenues to either academic or vocational training will be connected to the question of conscription. The point of view that military service is an agent of change is intended to secularize and acculturate Haredi youth into the Zionist narrative; this will remain at the core of Haredi discourse. At the same time, the degree to which the non-Haredi Jewish majority be willing to allow pathways to academic or vocational training without some kind of military service remains unclear.

**Funding:** This research was funded by a grant from the Tikvah Foundation.

**Institutional Review Board Statement:** Not applicable.

**Informed Consent Statement:** Not applicable.

**Conflicts of Interest:** The author declares no conflict of interest.

## Notes

[1]   (Mal'ach and Kahaner 2020).

[2]   (Friedman 1991).

[3]   For a discussion on poverty and Haredi community between 1950–2000 see: (Schueftan 2011, pp. 650–70).

[4]   Shenton, 42–46.

[5]   Sheaton, 13–14.

[6]   Sheaton, 15.

[7]   Shenaton, 25.

[8]   Shenaton, 24.

[9]   On the contemporary discourse on the issue see: (Malach 2014).

[10]  Shenaton, 30.

[11]  (Stadler 2002).

[12]  On the complex views concerning secular education and Haredi discourse see: (Mescheloff 2000; Bradney 2009).

[13]  https://iyun.org.il/article/%d7%97%d7%a8%d7%93%d7%99%d7%9d-%d7%9e%d7%94%d7%90%d7%a7%d7%93%d7%9e%d7%99%d7%94/%d7%91%d7%a2%d7%a0%d7%99%d7%99%d7%9f-%d7%a1%d7%9b%d7%a0%d7%aa-%d7%94%d7%90%d7%a7%d7%93%d7%9e%d7%99%d7%94/ (accessed on 4 June 2021).

[14]  https://iyun.org.il/article/%d7%97%d7%a8%d7%93%d7%99%d7%9d-%d7%9e%d7%94%d7%90%d7%a7%d7%93%d7%9e%d7%99%d7%94/%d7%91%d7%93%d7%99%d7%93%d7%95%d7%aa%d7%95-%d7%a9%d7%9c-%d7%94%d7%a1%d7%98%d7%95%d7%93%d7%a0%d7%98-%d7%94%d7%97%d7%a8%d7%93%d7%99/ (accessed on 4 June 2021).

[15]  (Barak-Koren 2019).

[16]  (Gonen 2000).

[17]  (Caplan 2006, p. 274).

[18]  (Caplan 2006, pp. 277–78).

[19]  (Roberts 2011; Continetti 2011).

20    (Caplan 2006, p. 278).

21    (Robinson 1998).

22    Concerning the role of the internet in transnationalism see: (Franz and Hiller 2004; Zaman 2008).

23    (Robinson 2004).

24    (Faist 2000, p. 4).

25    (Roudometof 2005, p. 68).

26    (Levi 1990).

27    Last accessed 15 May 2021 https://factfinder.census.gov/faces/tableservices/jsf/pages/productview.xhtml?pid=PEP_2017
       _PEPANNRES&prodType=table.

28    Last accessed 15 May 2021 https://factfinder.census.gov/faces/tableservices/jsf/pages/productview.xhtml?pid=ACS_16_5YR_
       S0101&prodType=table.

29    Last accessed 15 May 2021 https://factfinder.census.gov/faces/tableservices/jsf/pages/productview.xhtml?pid=ACS_16_5YR_
       B22005A&prodType=table.

30    Last accessed 15 May 2021 https://www.census.gov/quickfacts/fact/table/rocklandcountynewyork/EDU685216#viewtop.

31    Last accessed 15 May 2021 https://apps.mla.org/cgi-shl/docstudio/docs.pl?map_data_results.

32    The East Ramapo school district published information every year: https://www.ercsd.org/domain/54 (accessed on 1 November
       2023).

33    Interview with Menachem Berkowitz.

34    On the relationship between the Clintons and Skver see: Nick Anderson, Hasidic Clemency Case Entangles Hillary Clinton,
       Los-Angeles Times https://www.latimes.com/archives/la-xpm-2001-feb-24-mn-29756-story.html (accessed on 1 November
       2023). Yonit Shimron, "In voting, Orthodox Jews are looking more like evangelicals, Religious News Service, https://religionnews.
       com/2021/02/19/the-political-chasm-between-left-and-right-is-tearing-orthodox-jews-apart/ (accessed on 1 November 2023).

35    Marc Trencher, A Survey of Orthodox Jewish Political Attitudes and Behaviors: Haredi and Modern Orthodox Sectors. https://
       nishmaresearch.com/assets/pdf/REPORT%20-%20Orthodox%20Jewish%20Political%20Attitudes%20and%20Behaviors%20September%
       202023.pdf (accessed on 1 November 2023).

36    (Friedman 1991, pp. 10–47).

37    (Finkelman 2017).

38    (Heilman 2007).

39    For a discussion on the role of Lubavitch in the larger society see: (Feldman 2003).

40    (Mintz 1994).

41    (Itzhaki et al. 2018).

42    For a brief discussion on this issue see; (Hailman 2017).

43    (Lewitter 1982).

44    For a comparative view see: (Almog and Perry-Hazan 2011).

45    Interview with M.S.

46    Interview with S.P who teaches science at a local Litvish yeshiva.

47    see Note 45 above.

48    Interview with Y.T.

49    Interview with S.P.

50    See for example https://sarasch.com (accessed on 1 November 2023) or https://view2.fdu.edu/academics/petrocelli-college/
       academic-units/bais-yeshiva-program/ (accessed on 1 November 2023).

51    Interview with R.F.

52    See Note 51 above.

53    For a discussion of educational discourse on Education in Chabad see: (Solomon 2020).

54    See Note 51 above.

55    See Note 51 above.

56    https://mesivtamonsey.org (accessed on (accessed on 14 November 2023).

57    Interview with R.F.

58    Solomon, The Educational Philosophy, 182–192. The Chasidic discourse from 1978 was published in Sefer Ma'amarim Melukat,
       New-Yotrk, Kehot, in 1992, 57–68. This text references the rabbinic exegesis on the benefits of Torah versus "external wisdoms"
       (Hokhmot Hitzoniyot) and explains that these rabbinic texts do not mean to say that "external wisdoms" are worthless. They
       have value as long as they are used for holiness. The messianic era will see the elevation of this knowledge and how it can be
       commandeered for the sake of holiness. The unique argument in this text is that the elevation will not only be in the external
       knowledge but the Torah is also elevated in some aspect by this integration. These themes were repeated by R. Schneerson

many times through the years to explain the technological breakthroughs of the twentieth century, which he saw as a process that allowed for the dissemination of Godliness in more expansive ways. This is just one example of a topic in R. Schneerson's thought deserving of more attention using historical research methods.

59   See https://collive.com/college-announces-major-measure/ (accessed on (accessed on 1 November 2023 ).

60   Interview with M.S and Y.H.

61   (Barth et al. 2020).

62   https://www.mcl.org.il/ (accessed on 1 November 2023).

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
