# Peer review of "Changes in Haredi Education in Israel: A Comparative Perspective from the United States Using Monsey as a Test Case"

_religions, doi:10.3390/rel14111425_

Round 1

Reviewer 1 Report

Comments and Suggestions for Authors

The article is a thorough article and deals with one of the most pressing issues I the Israeli society – The Haredi community and its openness to academic education, both in Israel and in the US. There are two main problems with the article:

1.       It is very dense and full of information that is difficult for any scholar who is less familiar to the subject to follow. Terms like Haredi, Litvish, Lubavitch, Yeshiva, are sited without any attempt to explain to the reader what they are.

There is so much information about both communities in Israel and in the US, that I wonder if they don't deserve a separate article for each.

2.       Since the claim of the writer is to learn something from comparing both communities in Israel and in the US, a discussion about this discussion should be taken much more seriously. The writer mentions some challenges of the Israeli higher education, among them gender separation, but ignores the most important change for the Israeli society – the fact that most Haredi youth do not draft to the Israeli army.

The comparison between Israel and the US goes both ways. Israeli Haredi education influences US Haredi students as well. The writer chose to compare the whole Israeli Haredi community the community in Monsey, but although mentioning the community in Lakewood, which is by far less tolerant to academic education, I wonder how much was this community influenced by Israeli ideology.   

Author Response

Thank you for your comments. I made several changes and added to the body of the article and the footnotes to address these comments. 

  1. I added a definition to the term Haredi in the beginning of the article. This is based on the Israeli context. Under the title "The social framework of Monsey" there are definitions and references to the terminology mentioned in the first comment.
  2. I added paragraphs in several places referring to the influence on Lakewood originating in Israel and also referenced at the end of the article to the fact the question of army service will have an effect on any future change on Israeli Haredi education. 

Reviewer 2 Report

Comments and Suggestions for Authors

This is a very solid and compelling study. It is well written, lucid, well researched, and well structured. I enjoyed reading it and recommend it for publication. 

Author Response

Thanks you for your support and kind words. I have made a few changes based on the comments of the other reviewers. 

Reviewer 3 Report

Comments and Suggestions for Authors

The paper is important and deserves to be published. However, a number of issues deserve to be worked on to adjust it. I will list below.

1)      The comparative methodolgy adopted is not indicated in the title. The adopted theory must be indicated;

2)      In the fragmente “In recent years we have seen a growing scholarly interest in quanti-25 tative developments concerning one segment of the Jewish population, namely the Haredi community”. It should be cited which studies are these;

3)      In the introduction, reference is made to secularization. It is interesting to define its meaning for the paper;

4)      In the fragment “I am using Monsey as a test case 45 since it is a good representation”. It should indicate what the Monsey method ir. It is not clear;

5)      In the fragment “The sources show that the American Orthodox”, what sources are these?;

6)      It should explain more directly what the Haredi community is at the beginning of the paper;

Thus, the article deserves to be published, but first, it needs to be rewritten in the highlighted parts, in the attachment of theories, and footnotes along with the six points we have just indicated.

Author Response

Thank you for your comments I have integrated all of them in the revised version: 

  1. The term comparative is included in the title.
  2. I added a reference to Shnaton Ha-Hevrah Ha-Haredit.
  3. I have rewritten the section referring to Monsey as a test case and explained why it is a good comparison to Israel. 
  4. I changed the word sources to results. 
  5. I added several footnotes.